# Estimation of Urban–Rural Land Surface Temperature Difference at Different Elevations in the Qinling–Daba Mountains Using MODIS and the Random Forest Model

**DOI:** 10.3390/ijerph191811442

**Published:** 2022-09-11

**Authors:** Jiale Tang, Xincan Lan, Yuanyuan Lian, Fang Zhao, Tianqi Li

**Affiliations:** 1College of Geography and Environmental Science, Henan University, Kaifeng 475004, China; 2Key Laboratory of Geospatial Technology for the Middle and Lower Yellow River Regions (Henan University), Ministry of Education, Kaifeng 475004, China

**Keywords:** MODIS LST, DEM, land use, random forest, heat island effect, surface urban heat island, Qinling–Daba mountains

## Abstract

Land surface temperature (LST) variations are very complex in mountainous areas owing to highly heterogeneous terrain and varied environment, which complicates the surface urban heat island (SUHI) in mountain cities. Previous studies on the urban heat island (UHI) effect mostly focus on the flat terrain areas; there are few studies on the UHI effect in mountainous areas, especially on the influence of elevation on the SUHI effect. To determine the SUHI in the Qinling–Daba mountains (China), MODIS LST data were first preprocessed and converted to the same elevations (1500 m, 2000 m, 2500 m, 3000 m, and 3500 m) using a digital elevation model and the random forest method. Then, the average LSTs in urban land, rural land, and cultivated land were calculated separately based on the ranges of the invariable urban, rural, and cultivated areas during 2010–2018, and the urban, rural, and cultivated land LST difference were estimated for the same elevations. Results showed that the accuracy of LST estimated using the random forest method is very high (R^2^ ≥ 0.9) at elevations of 1500 m, 2000 m, 2500 m, 3000 m and 3500 m. The difference in urban, rural, and cultivated lands’ LST has a trend of decrease with increasing elevation, meaning that the SUHI weakens at higher elevations. The average LST of urban areas is 0.52–0.59 °C (0.42–0.57 °C) higher than that of rural and cultivated areas at an elevation of 1500 m (2000 m). The average LST of urban areas is 0.10–1.25 °C lower than that of rural and cultivated areas at elevations of 2500 m, 3000 m, and 3500 m, indicating absence of the SUHI at those elevations.

## 1. Introduction

Urbanization changes the characteristics of land use and land cover. One example is that of a natural surface dominated by vegetation, which will gradually transform into construction land with ongoing urban expansion [1,2,3]. Associated with that transition is the change of the underlying surface to an urban thermal environment that causes the temperature of a city to become higher than that of the surrounding suburbs, which is a phenomenon known as the UHI effect [4]. The UHI effect has an important role in the urban ecological environment and its overall livability [5]. The study of UHI effect and estimation of the urban heat island intensity (UHII) is therefore an important topic in various fields. As early as the beginning of the 19th century, Howard discovered the UHI and studied its impact on the urban climate of London (England) [6]. Subsequent studies calculated the intensity of the UHI and analyzed its spatiotemporal variations in Beijing [7], Nanjing [8], the Seattle area (USA) [9], New Jersey (USA) [10], and Turku, SW Finland [11], based on temperature records from different meteorological observation stations. Imhoff et al. used the surface permeable area of the Landsat dataset and MODIS land surface temperature (LST) data from 2003–2005 to conduct spatial analysis with the mean value of the annual cycle to study the UHI in the United States [12]. The combination of remote sensing and meteorological observation data makes it possible to explore the relationships between the variation of the UHI effect and land use change in cities with flat terrain [8]. However, it is difficult to determine the UHI effect for mountain cities owing to the relative lack of meteorological stations in mountainous areas and the complexity of LST associated with the highly heterogeneous terrain and varied environment [13,14,15,16]. In this study, the urban–rural surface temperature difference (including cultivated land) estimated by MODIS was used to quantitatively analyze the UHII in the Qinling–Daba mountains, and then explore the surface urban heat island (SUHI) in the Qinling–Daba mountains.

Mountain areas are characterized by complex temperature variation attributable to the underlying trend of decreasing temperature with increasing elevation and the mass–elevation effect [17,18,19,20], which increase the complexity of estimating the SUHI in mountain cities. The traditional method of estimating UHII is to compare the ground or air temperature observed at urban and rural meteorological stations [21,22,23]. However, this approach cannot be used to analyze the SUHI in mountainous regions because the urban and rural areas are distributed at different elevations. Only when the urban and rural temperatures are converted to the same elevations is it possible to determine the SUHI in mountainous areas. A standard temperature lapse rate of 0.6 °C/100 m has been used to determine temperatures at different elevations [24,25]. However, it produces certain errors in temperature estimation at the same elevations because the temperature lapse rate varies in different areas and different seasons, and because of the influence of the topography, geographical location, and underlying surface [26,27,28,29]. For example, the temperature lapse rate is 0.50 ± 0.02 °C/100 m on the northern flank of Taibai Mountain (China), but only 0.34 ± 0.04 °C/100 m on the southern flank [30]. It has been suggested that the random forest model and MODIS data could be used to convert temperatures to a unified elevation without considering the variation in temperature lapse rate [31]. Some scholars have shown that using the random forest method can generate rugged LST with high spatial and temporal resolution (R^2^ > 0.95 and RMSE around 3.00 K) in mountainous area with rugged terrain [32].

The Qinling–Daba mountains, located at the ecotone between subtropical and warm temperate zones in central China, are characterized by a complex environment and diversified altitudinal belts, namely the north–south transition zone [13,33,34]. The terrain is highly heterogeneous with elevations as high as 4000 m in the west and lower than 1000 m in the east [35]. To eliminate the influence of the error caused by elevation changes on SUHI, we first used MODIS LST data, DEM data and the random forest method to convert the LST to 1500 m, 2000 m, 2500 m, 3000 m and 3500 m, respectively. Then, we calculated the average LST in urban areas, rural areas, and cultivated land based on the ranges of the in-variable urban, rural, and cultivated land during 2010–2018, and the differences in urban–rural LST at the same elevations were analyzed to explore the SUHI in the Qinling–Daba mountains.

## 2. Materials and Methods

### 2.1. Data and Processing

#### 2.1.1. MODIS Data

MODIS raw data were downloaded from the NASA-Level-1 and Atmosphere Archive & Distribution System Distributed Active Archive Center-LAADS DAAC (Available online: https://ladsweb.modaps.Eosdis.nasa.gov/, accessed on 16 June 2020) from 2008 to 2018. The raw data of six MODIS products, including datasets and their spatial resolution, are shown in detail in Table 1. In this study, we preprocessed these MODIS data based on MRT (MODIS Reprojection Tool) with cropping based on the range of the Qinling–Daba mountains, projection transformation to WGS–84 and converting to TIFF format. All the data were averaged annually to obtain mean annual images from 2008 to 2018. NDWI, NDVI, ALB, LCT, and ET were resampled to 1-km resolution to remain consistent with LST.

NDWI can be calculated according to Formula (1):
$$\mathrm{NDWI}=\frac{R_{2}-R_{7}}{R_{2}+R_{7}}$$
where *R*_2_ and *R*_7_ are band 2 and band 7 data of MOD09A1 data products, respectively.

ALB were calculated based on the method proposed by Liang Shunlin [36] and MODIS surface reflectance band data according to the calculation Formula (2):ALB = 0.160α_1_ + 0.291α_2_ + 0.243α_3_ + 0.116α_4_ + 0.112α_5_ + 0.081α_7_ − 0.0015(2)
where α_1_, α_2_, α_3_, α_4_, α_5_ and α_7_ are band 1, band 2, band 3, band 4, band 5 and band 7 of MODIS09A1 products, respectively.

#### 2.1.2. DEM Data

DEM data used in this study were from the Space Shuttle Radar Topography Mission (STRM) with a spatial resolution of about 30 m and downloaded from the EarthExplorer-USGS website (Available online: https://earthexplorer.usgs.gov/, accessed on 6 February 2020). The raw data downloaded were spliced and cropped based on the range of the range of the Qinling–Daba mountains (Figure 1). We calculated Slope, Aspect, and mountain base elevation (MBE) based on a flow chart proposed by Liu [25], based on the DEM data and ArcGIS 10.3, then resampled them to 1-km resolution to be consistent with LST in the Qinling–Daba mountains. To obtain LST at the same elevations, LST raster datasets were reclassified according to the range of 1500 m, 2000 m, 2500 m, 3000 m and 3500 m in the DEM in the Qinling–Daba mountains.

#### 2.1.3. Land Use Data

As suburbs of urban areas mainly distributed cultivated land and rural land, LSTs of the urban land were compared with that of the rural land and cultivated land at elevations of 1500 m, 2000 m, 2500 m, 3000 m and 3500 m to explore the difference of urban–rural LST at the same elevations and explore the SUHI in the Qinling–Daba mountains. To obtain the range of the urban and rural areas that remained unchanged during 2010–2018, we downloaded land use data in 2010 and 2018 from the Resources and Environment Data Center of the Chinese Academy of Science (https://www.resdc.cn/ accessed on 17 May 2022) (Figure 2). Cultivated land, urban land and rural land were extracted from the original land use type classification system (25 secondary classifications and 6 primary classifications).

### 2.2. Random Forest Method

Random forest (RF) is an integrated learning method for classification, regression, and other tasks, proposed by Breiman in 2001. At present, random forest has been widely used in the scoring of variable importance [37], the downscaling of surface temperature [38,39], and to determine the relationship between surface temperature and other factors under complex terrain [28]. LST is an important parameter reflecting the surface environment, which regulates the air temperature in the lowest layer of the urban atmosphere and is also closely related to the nature of the subsurface and topographic conditions [15,40]. It was suggested that LST was affected by latitude (LAT), normalized difference water index (NDWI), normalized difference vegetation index (NDVI), evapotranspiration (ET), surface albedo (ALB), Slope and Aspect in mountainous areas [26]. With the increase in LAT, the solar radiation and the net radiation decrease greatly. LST therefore presents a trend of decreasing with the increase in LAT. NDWI has been shown to be related to LST [41,42,43]. NDVI is negatively correlated with LST and has the same seasonal variation characteristics as LST [44,45]. The change of ET and ALB affects the energy dynamic balance between land and air, and the latent heat flux of the surface, and therefore plays an essential role in the distribution of LST [46,47]. In mountain areas, the ground absorption of solar radiation is redistributed by Slope and Aspect, which affect the microclimate characteristics and variation of LST. Aside from that, mountain base elevation (MBE), as the average initial altitude of mountains or plateau in different parts, is suggested to be related to the warming effect of the mountain area [25,48], and is considered as one of the factors affecting LST. The change of LCT will alter ALB and ET, and the joint effect of ALB and ET may cause a warming effect, leading to the increase in LST, or cooling effect, leading to the decrease in LST [49].

In this study, we selected nine parameters (LAT, NDWI, NDVI, ET, LCT, ALB, Slope, Aspect, and MBE) as surface variables. Then, we established a nonlinear model between LST at different elevations and the surface variables based on the random forest method to explore the spatial variations of LST in the Qinling–Daba mountains. The nonlinear model is as follows:LST = f (LAT, NDWI, NDVI, ET, LCT, ALB, Slope, Aspect, MBE)(3)
where f is the nonlinear model established by the random forest method; LAT, NDWI, NDVI, ET, LCT, ALB, Slope, Aspect and MBE are parameters, LST is LST at elevations of 1500 m, 2000 m, 2500 m, 3000 m and 3500 m in the Qinling–Daba mountains. The estimated LST based on the random forest method was compared with the original LST at different elevations. R^2^ was calculated to evaluate the accuracy of the random forest model.

## 3. Results

### 3.1. Estimation of LST at the Same Elevations in the Qinling–Daba Mountains

#### 3.1.1. Accuracy Assessment of Estimating LST at the Same Elevations Based on the Random Forest Model

LSTs at elevations of 1500 m, 2000 m, 2500 m, 3000 m and 3500 m were estimated based on the method of random forest and nine variables (LAT, NDWI, NDVI, ET, LCT, ALB, Slope, Aspect and MBE) associated with LST and compared with the original LST at the same elevations. The results showed that estimated LST had an obvious trend of increasing as the original LST increased, and the R^2^ of linear regression models are as high as 0.9, 0.97, 0.98, 0.98, 0.97 at 1500 m, 2000 m, 2500 m, 3000 m and 3500 m (Figure 3 and Table 2). The RMSE are 0.79, 0.92, 0.93, 0.97 and 1,02 at 1500 m, 2000 m, 2500 m, 3000 m and 3500 m (Table 2).

#### 3.1.2. Variation of Estimated LST at the Same Elevations in the Qinling–Daba Mountains

The spatial variation of estimated LSTs at 1500 m, 2000 m, 2500 m, 3000 m and 3500 m in the Qinling–Daba mountains showed that the average LST reached 16.74 °C, 14.80 °C, 13.03 °C, 11.83 °C and 11.23 °C at 1500 m, 2000 m, 2500 m, 3000 m and 3500 m, respectively, in the Qinling–Daba mountains (Figure 4). Estimated LST had a trend of gradually increasing from east to west, from north to south, and from the edge of the mountain to the interior in the Qinling–Daba mountains. For example, the average LST of the grid passed by the east–west profile of 33° N was 16.81 °C, in the east (east of the 108° E) it was 10.88–21.43 °C and in the west (west of the 108° E) it was 10.24–23.95 °C at 1500 m; the average LST of the grid passed by south–north profile of 108° E was 16.35 °C, north of 33° N was 10.83–23.95 °C and south of 33° N was 10.24–23.50 °C at 1500 m.

### 3.2. Comparison of LST of Urban, Rural and Cultivated Land at Different Elevations

We extracted LST for the range of urban land, rural land and cultivated land that remained unchanged during 2010–2018 at five elevations of 1500 m, 2000 m, 2500 m, 3000 m, and 3500 m based on land use distribution in the Qinling–Daba mountains. LST presented a trend of decreasing from 1500 m to 3500 m at different regions, as shown in Table 3 and Figure 5. For example, LST was 18.07 °C, 17.55 °C and 17.48 °C, respectively, in urban land, rural land and cultivated land at 1500 m, but they decreased to 10.18 °C, 11.01 °C and 11.43 °C, respectively, at 3500 m. LST was higher in the urban areas than that in the rural land and cultivated land at elevations of 1500 m and 2000 m, while from 2500–3500 m, LST was lower in the urban areas than that in the rural land and cultivated land.

The average differences of urban–rural LST were calculated based on LST in the urban, rural, and cultivated land at different elevations. It was shown that the higher the elevation, the lower the difference of urban–rural LST. LST was 0.52–0.59 °C higher in urban land than that rural land and cultivated land at 1500 m, but the difference between urban land and rural land, cultivated land LST decreased to 0.42–0.57 °C at 2000 m (Table 4, Figure 6).

The average LST of urban land is less than that of the rural and cultivated lands at elevations of 2500 m and above, meaning SUHI has disappeared in the middle and alpine mountains in the Qinling–Daba mountains. The higher the elevation, the smaller the urban land than the rural and cultivated lands. The average LST was 0.10 and 0.33 °C lower in the urban land than in the rural land and cultivated land at 2500 m; they increased to 0.37 and 0.71 °C lower in the former than in the latter at 3000 m, and then to 0.83 and 1.25 °C lower in the former than in the latter at 3500 m in the Qinling–Daba mountains (Figure 6).

## 4. Discussion

The results showed that the SUHI had a trend of decrease with increasing elevation in the mountains of the study area. The LST was 0.52–0.59 °C higher in urban land than in rural land and cultivated land at 1500 m, but the difference between urban land and rural land, cultivated land LST decreased to 0.42–0.57 °C at 2000 m. The SUHI almost disappeared at elevations of 2500–3500 m. The possible reason for this elevation–related phenomenon of the SUHI is the mass–elevation effect in mountainous areas. It has been shown that the mass–elevation effect has a positive impact on temperatures, which leads to a warming effect that is positively correlated with elevation in mountainous areas [24,25,50,51,52]. The warming effect caused by the mass–elevation effect for rural land and cultivated land in middle and alpine regions (2500–3500 m) neutralized or exceeded the SUHI of urban land, resulting in disappearance of the SUHI at higher elevations. This result explains the variation of the SUHI with elevation in mountainous areas, and to a certain extent improves the understanding of the variation of the SUHI at different elevations. The mass–elevation effect and the SUHI interacted make the variation of LST very complex in mountain areas.

Many previous studies focused on comparison of air temperatures from urban meteorological stations and rural meteorological stations to explore the characteristics of the UHI, and to reveal the differences between urban and rural areas [7,8,9]. In this study, we found that the UHI effect in mountainous areas plays an important role in the variation of LST, causing the LST to be 0.42–0.59 °C higher for urban land than for both rural and cultivated land below 2000 m. Because LST is influenced by the properties of the underlying surface and terrain conditions [15,40], it has smaller variation than air temperature [15]. Therefore, the UHII estimated by the LST is slightly lower than that based on air temperatures. For example, Gantuya Ganbat et al. [23] and Yeon-Hee Kim et al. [53] reported that the average UHII in the study region is 1.6 and 2.2 °C, respectively, i.e., approximately 1.0–1.6 °C higher than our results.

Urban scope has a trend of expansion owing to increasing urbanization, especially during the previous 20 years [34], and the change of urban areas has an important influence on UHI effect [54]. To estimate the spatial differences in LST between urban and rural areas, we selected the unaltered ranges of urban, rural, and cultivated land during 2010–2018 to explore the SUHI in the Qinling–Daba mountains. Although rural areas were composed of cultivated land, forest land and grassland and rural land, this study compared the LST difference between the urban, rural, and cultivated land at the same elevations to explore the SUHI in the Qinling–Daba mountains. That is because forest land and grassland have greater impacts on climate by themselves, such as the impact of forest land on carbon sequestration, leading to climate cooling [55], and the great impact of grassland on surface albedo [34]. However, to some extent, this ignored the SUHI caused by newly added urban areas during 2010–2018. In future study, the temporal variation of the SUHI should be explored with consideration of the continuous expansion of the urban areas. Remote sensing products with higher resolution could be used to determine the SUHI at the local scale, although the 1-km resolution MODIS data are adequate for determining the SUHI in the Qinling–Daba mountains, which cover an area of approximately 300,000 km^2^. Moreover, information on meteorological stations and local climate factors (e.g., monsoon and urbanization processes) could be integrated to reveal the spatiotemporal variations and analyze the influencing factors of the UHI effect, to provide reliable reference suggestions for promoting the sustainable development of cities in the future and improving the quality of life of residents. In the future, we will identify the boundaries between cities and suburbs on a smaller scale and explore the UHI and SUHI on a smaller scale.

## 5. Conclusions

(1)The random forest method and MODIS data can be used to estimate LST at standard elevations. It was shown that the LST estimated using MODIS and the random forest method has obvious linear correlation with the original LST, with R^2^ values of >0.9 at elevations of 1500 m, 2000 m, 2500 m, 3000 m and 3500 m.(2)The difference in urban–rural LST has a trend of decrease with increasing elevation, meaning that the SUHI tends to weaken at higher elevations. The average LST of urban areas is 0.52–0.59 °C higher than that of rural and cultivated lands at 1500 m, but the former is 0.42–0.57 °C higher than the latter at the elevation of 2000 m.(3)The average LST of urban areas is less than that of rural areas at elevations of ≥2500 m, meaning that the SUHI disappears in the middle and alpine mountains in the Qinling–Daba mountains. The average LST of urban areas is 0.1 °C, 0.37 °C, and 0.83 °C (0.33 °C, 0.71 °C and 1.25 °C) lower than that of rural and cultivated lands at elevations of 2500 m, 3000 m, and 3500 m, respectively.

## Figures and Tables

**Figure 1 ijerph-19-11442-f001:**
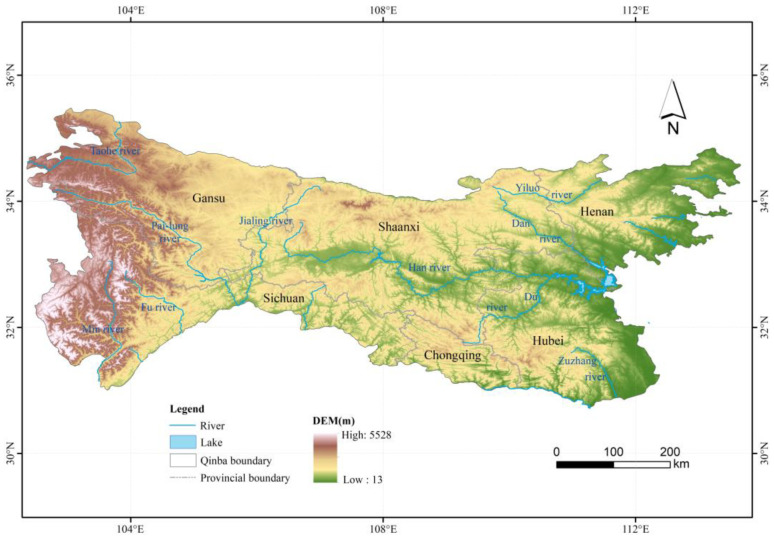
Map showing elevation of the Qinling–Daba mountains.

**Figure 2 ijerph-19-11442-f002:**
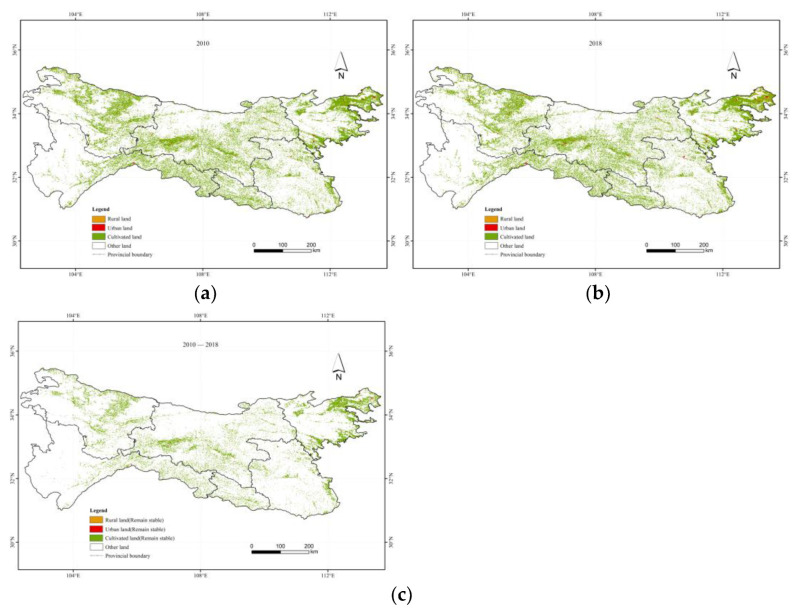
Land use in the Qinling–Daba mountains after reclassification to four types: cultivated land, urban land, and rural land in (**a**) 2010 and (**b**) 2018. (**c**) Extraction of cultivated land, urban land, and rural land that remained unchanged during 2010–2018 in the Qinling–Daba mountains.

**Figure 3 ijerph-19-11442-f003:**
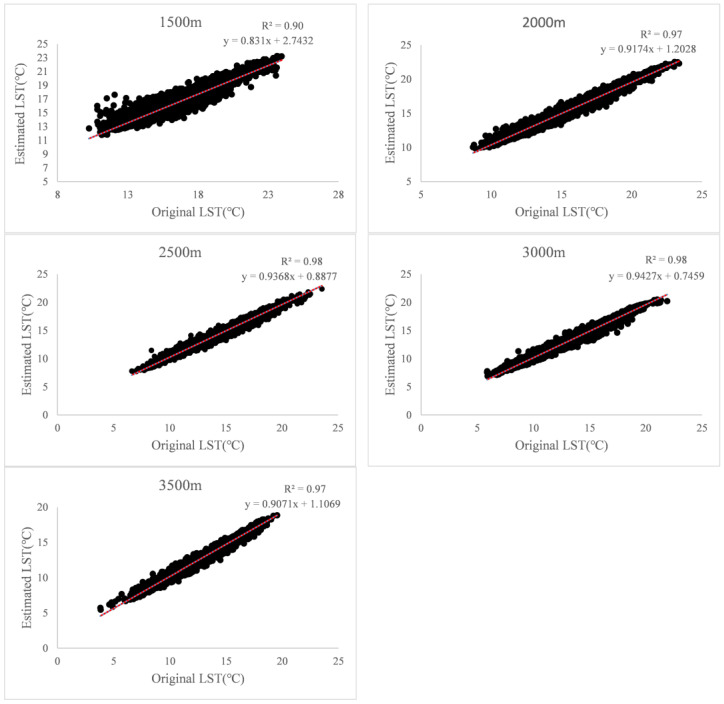
Comparison of LST estimated using the random forest model and nine variables (LAT, NDWI, NDVI, ET, LCT, ALB, Slope, Aspect, and MBE) and original LST at elevations of 1500 m, 2000 m, 2500 m, 3000 m, and 3500 m.

**Figure 4 ijerph-19-11442-f004:**
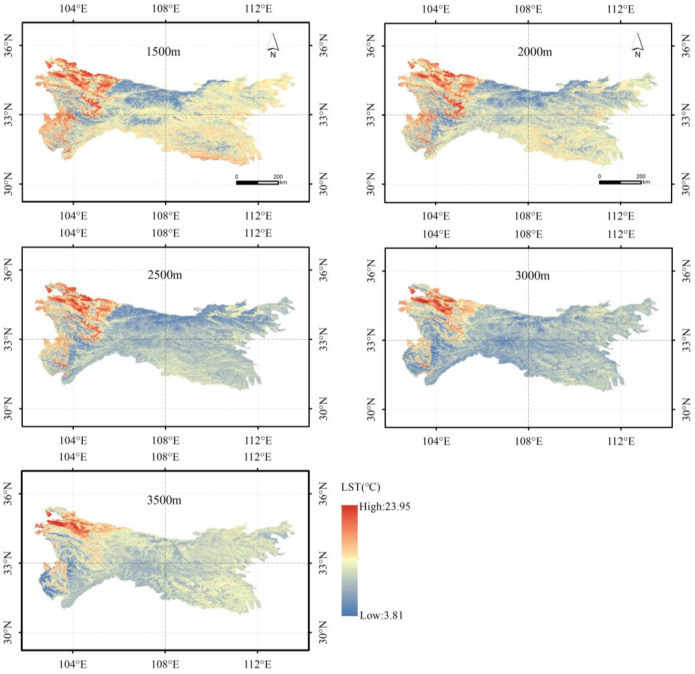
Spatial variation of LST at 1500 m, 2000 m, 2500 m, 3000 m and 3500 m in the Qinling–Daba mountains estimated using the random forest model.

**Figure 5 ijerph-19-11442-f005:**
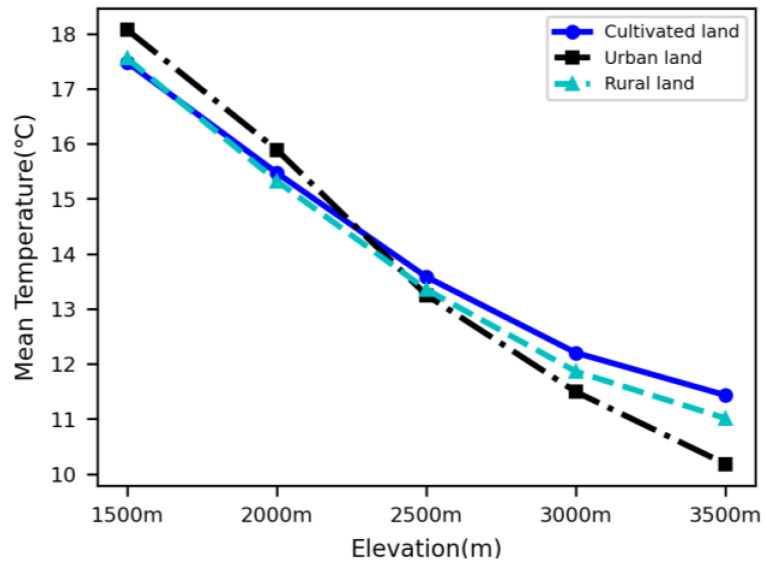
Variations of LST for urban, rural, and cultivated land with elevation.

**Figure 6 ijerph-19-11442-f006:**
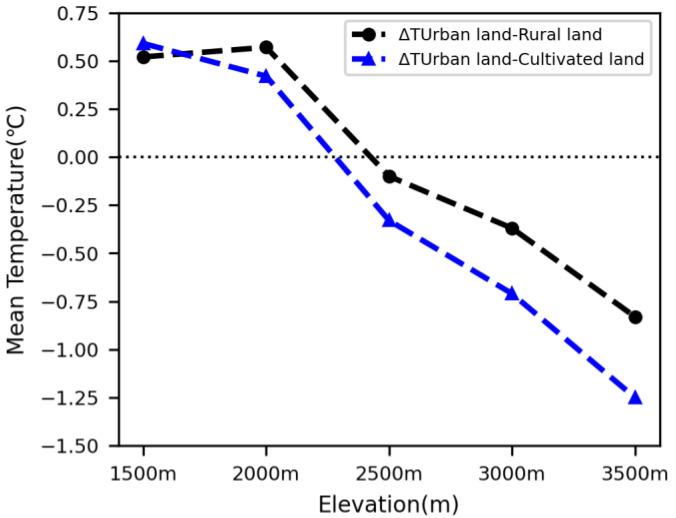
Variations of LST (°C) differences between urban land and other land use types with elevation.

**Table 1 ijerph-19-11442-t001:** MODIS products and related variables used in this study.

Variable	Datasets	Temporal Resolution	Spatial Resolution
Land surface temperature (LST)	MOD11A1	8 Day	1 km
Normalized difference water index (NDWI)	MOD09A1	8 Day	500 m
Normalized difference vegetation index (NDVI)	MOD13A2	16 Day	1 km
Surface albedo (ALB)	MOD09A1	8 Day	500 m
Land cover type (LCT)	MCD12Q1	Year	500 m
Evapotranspiration (ET)	MOD16A2	8 Day	500 m

**Table 2 ijerph-19-11442-t002:** Accuracy and error of LST estimated using the random forest model at the same elevations.

Statistical Indicators	1500 m	2000 m	2500 m	3000 m	3500 m
R^2^	0.90	0.97	0.98	0.98	0.97
RMSE	0.79	0.92	0.93	0.97	1.02

**Table 3 ijerph-19-11442-t003:** Comparison of LST (°C) of urban, rural, and cultivated land at elevations of 1500 m, 2000 m, 2500 m, 3000 m, and 3500 m.

	1500 m	2000 m	2500 m	3000 m	3500 m
Urban land	18.07	15.89	13.25	11.49	10.18
Rural land	17.55	15.32	13.35	11.86	11.01
Cultivated land	17.48	15.47	13.58	12.2	11.43

**Table 4 ijerph-19-11442-t004:** Temperature differences between LST (°C) of urban land and other land use types (rural and cultivated land) at different elevations.

	1500 m	2000 m	2500 m	3000 m	3500 m
ΔTUrban land–Rural land	0.52	0.57	−0.10	−0.37	−0.83
ΔTUrban land–Cultivated land	0.59	0.42	−0.33	−0.71	−1.25

## Data Availability

The data that support the findings of this study are available from the authors upon reasonable request.

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
