# Peer review of "Estimation of Urban–Rural Land Surface Temperature Difference at Different Elevations in the Qinling–Daba Mountains Using MODIS and the Random Forest Model"

_ijerph, 2022, doi:10.3390/ijerph191811442_

Round 1

Reviewer 1 Report

This study estimated the UHI intensity at the different elevations in Qinling-daba mountains. Generally, this study is useful to improve the understanding of UHII at different elevations. While this manuscript need much improvements.  

Line 35: Estimation of UHI intensity? Effect includes a lot of things, in this study, I think you focused on UHI intensity. And the UHI based on surface temperature is SUHI, different from the traditional UHI based on air temperature.

Line 64 to 66, can you give some examples based on random forest and the accuracy of the converted LST? e.g. Ouyang, X.; Dou, Y.; Yang, J.; Chen, X.; Wen, J. High Spatiotemporal Rugged Land Surface Temperature Downscaling over Saihanba Forest Park, China. Remote Sens. 2022, 14, 2617. https://doi.org/10.3390/rs14112617

Line 73: grammar error. And there are too much long sentences, which area very confused. I suggest authors check the grammar carefully and rewrite long sentences.

Figure 2: I cannot find out the area of urban areas in this figure, can you make it clearer?

Line 167: how about the RMSE between the estimated and original LST?

In discussion section, the significance and the implication of this study should be presented. Additionally, the limitation of this study should be mentioned.

Author Response

Response to Reviewer 1 Comments

We gratefully thank the editor and all reviewers for their constructive remarks and suggestions, which has significantly improved the manuscript. All the comments proposed by the reviewers were responded point by point and the revisions were indicated. All the revisions to the manuscript have been marked using the "Track Changes" function.

General Comments: This study estimated the UHI intensity at the different elevations in Qinling-daba mountains. Generally, this study is useful to improve the understanding of UHII at different elevations. While this manuscript need much improvements.  

Point 1: Line 35: Estimation of UHI intensity? Effect includes a lot of things, in this study, I think you focused on UHI intensity. And the UHI based on surface temperature is SUHI, different from the traditional UHI based on air temperature.

Response 1: Thank you for the comment. The UHI has been corrected to SUHI in this paper. The surface temperature difference between urban and rural areas (including cultivated land) estimated based on MODIS is used to quantify the intensity of UHI in the Qinling-Daba mountains. The SUHI is more appropriate and we have changed UHI to SUHI in the revised manuscript.

Point 2: Line 64 to 66, can you give some examples based on random forest and the accuracy of the converted LST? e.g. Ouyang, X.; Dou, Y.; Yang, J.; Chen, X.; Wen, J. High Spatiotemporal Rugged Land Surface Temperature Downscaling over Saihanba Forest Park, China. Remote Sens. 2022, 14, 2617. https://doi.org/10.3390/rs14112617.

Response 2: Thank you for your valuable suggestion. We have cited the literature you recommended in the proper place of the revised manuscript.

Point 3: Line 73: grammar error. And there are too much long sentences, which area very confused. I suggest authors check the grammar carefully and rewrite long sentences.

Response 3: Thank you for the suggestion. We checked the sentences and grammar carefully. The grammar errors have been corrected and some long sentences have been changed to short sentences in the revised manuscript.

Point 4: Figure 2: I cannot find out the area of urban areas in this figure, can you make it clearer?

Response 4: Thank you for the comment. In this paper, we selected the unaltered ranges of urban, rural, and cultivated land from 2010 to 2018 to estimate the land surface temperature difference between urban and rural areas and explore the SUHI effect in the Qinling–Daba mountains. However, the terrain of the Qinling–Daba mountains was very rugged and the land use is dominated by woodland, the urban and rural areas were relatively small and fragmented. More striking red, yellow and green were used to show urban, rural, and cultivated land to make them identified easier.

Point 5: Line 167: how about the RMSE between the estimated and original LST?

Response 5: Thank you for the comment. We added Table 2 in the revised manuscript and added RMSE between estimated LST and original LST to Table 2, so as to facilitate a clearer view of statistical indicators.

Table 2. Accuracy and error of LST estimated using the random forest model at the same elevations.

Statistical indicators

1500m

2000m

2500m

3000m

3500m

0.90

0.97

0.98

0.98

0.97

RMSE

0.79

0.92

0.93

0.97

1.02

Point 6: In discussion section, the significance and the implication of this study should be presented. Additionally, the limitation of this study should be mentioned.

Response 6:Thank you for your comment. We have appropriately added an explanation of the significance of this study in the Discussion section. The limitations of this study was described in the third paragraph of the Discussion section.

Reviewer 2 Report

Still corrections are required.

references are still not updated.

The Methodology section is required to be reframed.

Author Response

Response to Reviewer 2 Comments

We gratefully thank the editor and all reviewers for their constructive remarks and suggestions, which has significantly improved the manuscript. All the comments proposed by the reviewers were responded point by point and the revisions were indicated. All the revisions to the manuscript have been marked using the "Track Changes" function.

General Comments: Still corrections are required.

references are still not updated.

The Methodology section is required to be reframed.

Response: Thank you for your comments. We have made relevant revisions to the article, and any revision to the manuscript has been marked using the "Track Changes" function. We have added several more relevant references to support our research. In the method section, random forest method and its application were described in detail first, and then we described LST and its influencing factors. We used the random forest method and nine LST-related variables to design a nonlinear link model to estimate the surface temperature at same altitudes to estimate the surface temperature difference between urban and rural areas (including cultivated land) to explore the SUHI (surface urban heat island) effect in the Qinling–Daba Mountains.

Reviewer 3 Report

The article studies the urban heat island effect, land surface temperature`s (LST) variation and their complexity in mountain cities. To study the influence of elevation on LST and to determine the UHI effect in the Qinling–Daba mountains, Modis LST data were converted to the same elevation using a digital elevation model and the random forest method. Then, the average LST in urban and rural areas was calculated separately based on the ranges of the invariable urban and rural areas, and the urban–rural LST difference was estimated for the same elevation. Results showed that the accuracy of LST estimated using the random forest method is very high (R2 ≥ 0.9) at elevations of 1500, 2000, 2500, 3000, and 3500 m. The difference in urban–rural LST has a trend of decrease with increasing elevation, meaning that the UHI effect weakens at higher elevations.

 Suggestions:

Please, notice that:

1. You have citations 7, 8, 10 and 11. 

UHI effect has been studied also in Nordic conditions: Suomi, J. (2014). Characteristics of urban heat island (UHI) in a high-latitude coastal city: A case study of Turku, SW Finland. University of Turku.

2. List of abbreviations would be a useful addition for reader.

Author Response

Response to Reviewer 3 Comments

We gratefully thank the editor and all reviewers for their constructive remarks and suggestions, which has significantly improved the manuscript. All the comments proposed by the reviewers were responded point by point and the revisions were indicated. All the revisions to the manuscript have been marked using the "Track Changes" function.

General Comments: The article studies the urban heat island effect, land surface temperature`s (LST) variation and their complexity in mountain cities. To study the influence of elevation on LST and to determine the UHI effect in the Qinling–Daba mountains, Modis LST data were converted to the same elevation using a digital elevation model and the random forest method. Then, the average LST in urban and rural areas was calculated separately based on the ranges of the invariable urban and rural areas, and the urban–rural LST difference was estimated for the same elevation. Results showed that the accuracy of LST estimated using the random forest method is very high (R2 ≥ 0.9) at elevations of 1500, 2000, 2500, 3000, and 3500 m. The difference in urban–rural LST has a trend of decrease with increasing elevation, meaning that the UHI effect weakens at higher elevations.

Point 1:1. You have citations 7, 8, 10 and 11.  UHI effect has been studied also in Nordic conditions: Suomi, J. (2014). Characteristics of urban heat island (UHI) in a high-latitude coastal city: A case study of Turku, SW Finland. University of Turku.

Response 1:Thank you for your valuable suggestion. We have cited the literature you recommended in the revised manuscript.

Point 2 : List of abbreviations would be a useful addition for reader.

Response 2: Thank you for your valuable suggestion. In the methods section we have refined the full name of each acronym.

“It was suggested that LST was affected by Latitude (LAT), Normalized difference water index (NDWI), Normalized difference vegetation index (NDVI), Evapotranspiration (ET), Surface albedo (ALB), Slope and Aspect in mountainous areas [23]. Aside from that, Mountain base elevation (MBE) [40],Land cover type (LCT) [35]can also affect LST.”

Round 2

Reviewer 2 Report

All issues has been addressed

Author Response

Response to Reviewer Comments

We gratefully thank the editor and all reviewers for their constructive remarks and suggestions, which has significantly improved the manuscript. All the comments proposed by the reviewers were responded point by point and the revisions were indicated. All the revisions to the manuscript have been marked using the "Track Changes" function.

General Comments:All issues has been addressed.

Response: Thank you for your comments. We have made relevant revisions to the article, and any revision to the manuscript has been marked using the "Track Changes" function. We have made relevant supplements in the Abstract and Discussion sections of the manuscript to explanations on the shortcoming of recent studies and to provide perspectives for further in-depth research. We have carefully checked and revised the English language, style and spelling of words in our manuscript.
